# Polygenic Risk Score in Predicting Esophageal, Oropharyngeal, and Hypopharynx Cancer Risk among Taiwanese Population

**DOI:** 10.3390/cancers16040707

**Published:** 2024-02-07

**Authors:** Yu-Che Huang, Ming-Ching Lee, Sheng-Yang Huang, Chia-Man Chou, Hui-Wen Yang, I-Chieh Chen

**Affiliations:** 1Division of Thoracic Surgery, Department of Surgery, Taichung Veterans General Hospital, Taichung 40705, Taiwan; berkiekoyukih@gmail.com; 2Department of Critical Care Medicine, Taichung Veterans General Hospital, Taichung 40705, Taiwan; 3Department of Medical Education, Taichung Veterans General Hospital, Taichung 40705, Taiwan; 4Division of Pediatric Surgery, Department of Surgery, Taichung Veterans General Hospital, Taichung 40705, Taiwan; drugholic@vghtc.gov.tw (S.-Y.H.); cmchou@mail.vghtc.gov.tw (C.-M.C.); 5Department of Post-Baccalaureate Medicine, College of Medicine, National Chung Hsing University, Taichung 40227, Taiwan; 6School of Medicine, National Yang Ming Chiao Tung University, Taipei 11267, Taiwan; 7Department of Medical Research, Taichung Veterans General Hospital, Taichung 40705, Taiwan; wendy210001@gmail.com

**Keywords:** esophageal cancer, oropharyngeal cancer, hypopharyngeal cancer, polygenic risk score, incidence

## Abstract

**Simple Summary:**

In this retrospective study conducted at Taichung Veterans General Hospital, 54,692 participants were examined, including 385 with esophageal, oropharyngeal, or hypopharyngeal squamous cell carcinoma (SCC). Researchers investigated the correlation between cancer incidence and prognosis and a polygenic risk score (PRS) derived from 8353 single-nucleotide polymorphisms. Those with a high PRS (Q4) exhibited a 1.83-fold higher risk of SCCs compared to the low-PRS group (Q1), particularly notable in esophageal and hypopharyngeal cancers. Notably, PGS001087 displayed discernible associations with SCC risk, especially in specific subtypes and advanced stages, although not significantly linked to clinical staging. The study underscores the significance of genetic factors in upper aerodigestive tract cancers, notably esophageal SCC, offering insights for future research and risk assessment strategies. These findings highlight the importance of PRS in understanding cancer susceptibility, guiding targeted interventions, and informing personalized treatment approaches.

**Abstract:**

Esophageal cancer shares strong associations with oropharyngeal and hypopharyngeal cancers, primarily due to shared risk factors like excessive tobacco and alcohol use. This retrospective study at Taichung Veterans General Hospital involved 54,692 participants, including 385 with squamous cell carcinoma (SCC) of the esophagus, oropharynx, or hypopharynx. Using a polygenic risk score (PRS) derived from 8353 single-nucleotide polymorphisms, researchers aimed to assess its correlation with cancer incidence and prognosis. The study found a 1.83-fold higher risk of esophageal, oropharyngeal, and hypopharyngeal SCCs in participants with a high PRS (Q4) compared to the low-PRS group (Q1). Esophageal cancer risk demonstrated a significant positive association with the PRS, as did hypopharyngeal cancer. Clinical parameters and staging showed limited associations with PRS quartiles, and the PRS did not significantly impact recurrence or mortality rates. The research highlighted that a higher PRS is linked to increased susceptibility to esophageal and hypopharyngeal cancer. Notably, a specific polygenic risk score, PGS001087, exhibited a discernible association with SCC risk, particularly in specific subtypes and advanced disease stages. However, it was not significantly linked to clinical cancer staging, emphasizing the multifactorial nature of cancer development. This hospital study reveals that a higher PRS correlates with increased susceptibility to esophageal and hypopharyngeal cancers. Notably, PGS001087 shows a discernible association with SCC risk in specific subtypes and advanced stages, although not significantly linked to clinical cancer staging. These findings enhance our understanding of genetic factors in upper aerodigestive tract cancers, particularly esophageal SCC, guiding future research and risk assessment strategies.

## 1. Introduction

Esophageal cancer has been considered closely related to oropharyngeal cancer and hypopharyngeal cancer as they are all commonly caused by excessive tobacco and alcohol consumption and they may occur as synchronous or metachronous double cancers [1]. In Taiwan, in 2020, there were 1689, 1170, and 2875 newly diagnosed cases of oropharyngeal cancer, hypopharyngeal cancer, and esophageal cancer, respectively. Furthermore, there were 507, 692, and 1954 deaths due to oropharyngeal cancer, hypopharyngeal cancer, and esophageal cancer, respectively [2]. Moreover, the 5-year survival rate for locally or systemically advanced patients could be lower than 10–20% [3,4].

Previous studies have identified several common genetic polymorphisms that are associated with the risk of developing esophageal cancer [3]. Recently, the concept of the polygenic risk score (PRS) has been introduced and has been considered as a composite score that incorporates the presence or absence of multiple genetic variants associated with particular cancers [5]. It has begun to be used as a prediction factor for different cancers, such as prostate [6], lung [7], breast [8], and colorectal cancer [9].

More PRS studies into cancers of the upper gastrointestinal tract have also been published. In 2020, Jin et al. found PRSs reliable in predicting the risk of gastric cancer, which could be reduced via lifestyle adjustment [10]. This research highlighted the potential for risk reduction through lifestyle adjustments, underlining the importance of early identification and intervention. In February 2023, Wang et al. also found that PRSs had good potential in the prediction of gastric cancer, with better prediction accuracy with the incorporation of epidemiological factors or *Helicobacter pylori* (*H. pylori*) status [11]. Furthermore, He et al. discovered that PRSs significantly enhances risk stratification for nasopharyngeal carcinoma (NPC) and contributes to personalized screening, particularly when combined with Epstein–Barr virus (EBV) testing [12], highlighting the pivotal role of PRSs in enhancing risk stratification for NPC and shaping personalized screening strategies.

The current published PRSs for esophageal cancer have been developed using data from individuals of European ancestry. While these studies have made notable progress in the field of cancer risk prediction, it is worth noting that PRS-related research for cancers of the esophagus, oropharynx, and hypopharynx remains relatively limited, especially in a Taiwanese population. Moreover, as a common abused substance, alcohol has already been confirmed a major risk factor for SCCs of the upper gastrointestinal tract [13,14]. Kumagai et al. even proved that the amount and time period of alcohol consumption may increase the risk of esophageal SCC [15]. However, until now, there has been no published PRS utilized yet to evaluate the correlation between alcohol consumption and alcohol-related cancers in Asian populations. Therefore, the aim of our study is to investigate the association between the PRS and susceptibility to SCCs of the esophagus, oropharynx, and hypopharynx, as well as to analyze the impact of genome-wide susceptibility variants on clinical outcomes and prognosis.

## 2. Patients and Methods

### 2.1. Study Design and Study Population

This retrospective case–control study was conducted within a hospital setting and encompassed 57,257 Taiwanese participants. The study drew upon data from the Taiwan Precision Medicine Initiative (TPMI) project, overseen by Academia Sinica in Taiwan, and was carried out from June 2019 to November 2022. Participant data were collected from Taichung Veterans General Hospital (TCVGH) over the period spanning from April 2009 to December 2021. We enrolled participants aged ≥20 years in partnership with the TPMI project supervised by Academia Sinica, Taiwan from TCVGH. Detailed information of participants including demographic data, medical records, physical examinations and blood tests were collected, and all received genotyping using the Affymetrix Genome-Wide TWB 2.0 SNP Array. The study was approved by the ethics committee of TCVGH Institutional Review Board (IRB No. CE23510B), and all participants provided informed consent. Clinical parameters were obtained from the dataset and electronic medical records from TCVGH using a de-identification method, and all of the participants provided written informed consent, which was obtained in accordance with the principles defined in the Declaration of Helsinki.

### 2.2. Study Participants

This study analyzed data from a total of 57,257 participants in the TCVGH database. After excluding 2295 participants due to missing data, the final dataset comprised 54,962 participants in the TCVGH database. Out of all the participants, 191 patients were clinically diagnosed with esophageal cancer, as confirmed by the International Classification of Diseases, Clinical Modification (ICD-9-CM) diagnosis codes 150.x. Additionally, 135 patients were clinically diagnosed with oropharyngeal cancer, confirmed by the ICD-9 cm 146.6–146.9, and 110 patients were clinically diagnosed with hypopharyngeal cancer, confirmed by the ICD-9 cm 148.2–148.9. These diagnoses were either recorded at least twice during outpatient visits or once during hospitalization. The index date was defined as the date of their cancer diagnosis.

### 2.3. Genotyping, Imputation, and PRSs

The blood DNA samples were extracted from the participants, and genotyping was performed using the Axiom Genome-Wide TWB 2.0 (TWBv2.0) Array Plate (Affymetrix, Santa Clara, CA, USA), which included 114,000 risk variants in 2831 unusual disease genes selected from ClinVar, ACMG, GWAS Catalog, HGMD, locus-specific databases, and the published literature [16]. Genetic variants were aligned to the human genome reference GRCh38. The genotyping chip used in this study was specifically tailored for the Han Chinese population in Taiwan, encompassing a total of 714,431 SNPs [17]. Rigorous data analysis and quality control procedures were applied using Affymetrix Power Tools. Markers related to sex hormones were excluded, along with those exhibiting a minor allele frequency of less than 0.01 or a genotype missing rate exceeding 5% [16]. Furthermore, markers failing the Hardy–Weinberg equilibrium tests with a *p*-value lower than 1.0 × 10^−5^ were removed [16]. Genotype imputation was carried out across the autosomal chromosomes using the Michigan Imputation Server, which implemented the ‘minimac4′ algorithm [18]. Strand-aligned genotype data were loaded into the server. We performed the imputation using the 1000 Genomes Phase 3 (Version 5) reference panel [19]. All biallelic variants with an imputation quality threshold of INFO score ≥ 0.3 were reported.

In this study, we utilized the PGS001087 PRS and PGS001394, which were derived from a discovery analysis of variants associated with alcohol consumption measurement identified from a trans-ancestry genome-wide association study (GWAS) for 8353 and 860 curated traits, respectively, across nine ancestry groups from the United Kingdom Biobank [20].

We obtained the list of SNPs and their corresponding effect sizes from the Polygenic score catalog. The PRS was calculated by using the ‘score’ function from plink version 1.9 [21] to aggregate the effects of multiple genetic variants weighted by their effects size from the polygenic score catalog [22]. We used the approach is to weight each allele dosage by its effect size, as described previously [23].

Equation: standard equation to calculate a weighted polygenic risk score:$${PRS}_{j}=\sum_{i}^{N}\beta_{i}\times {dosage}_{ij}$$

*N* represents the total number of SNPs in the polygenic score.

$\beta_{i}$ represents the effect size (often denoted as beta) associated with variant *i*.

${dosage}_{ij}$ stands for the number of copies of SNP *i* present in the genotype of individual *j*.

We divided the 54,692 TPMI participants into four equal groups based on their PRS from highest to lowest. The number of individuals in each group was the same. In this study, participants were grouped into Q1 (0–25%), Q2 (26–50%), Q3 (51–75%), and Q4 (76–100%) of scores according to these thresholds.

### 2.4. Clinical Parameters

Patients diagnosed with esophagus, oropharynx, and hypopharynx cancer were identified based on International Classification of Disease, Ninth Revision (ICD-9-CM) code 150.x, 146.6–146.9, and 148.2–148.9, respectively, which required positive pathological proof through endoscopic biopsy during the period from January 2009 to January 2022. All participants with upper aerodigestive tract cancers were incident cases and had not undergone regular follow-up prior to their cancer diagnoses. The genetic profile was correlated with clinical parameters such as age at onset, sex, family history of esophagus, oropharynx, and hypopharynx cancer, history of smoking, alcohol drinking and betel nut chewing, squamous cell carcinoma antigen (SCC) levels, and clinical stages. Cigarette smoking, alcohol drinking, and betel nut chewing were ascertained through self-reports from the upper aerodigestive tract cancer patients and defined as “yes” or “no”. Outcome evaluations included overall mortality and local or distant recurrence, which was proven via positron emission tomography (PET) scan or positive pathological findings followed by endoscopic biopsy.

### 2.5. Statistics Analyses

Hazard ratios (HRs) were calculated using Cox proportional hazard regression models, stratified by sex, and using time since study entry as the timescale. Outcomes were censored if a participant was lost to follow-up or died, or if the end of available follow-up was reached (November 2022). Descriptive statistics for continuous variables were presented as mean ± standard deviation (SD), and group differences were assessed using Student’s *t*-test. Categorical variables were expressed as number (percent), and differences among groups were evaluated using the Chi-square test and, in cases where the minimum expected count was less than 5, we utilized Fisher’s exact test. Two-tailed statistical tests were employed, with significance considered at *p* < 0.05. All analyses were carried out using IBM SPSS Statistics package, version 25 (IBM Corporation, Armonk, NY, USA), and SAS, version 9.4 statistical software (SAS Institute Inc., Cary, NC, USA).

## 3. Results

This study analyzed data from a total of 57,257 participants in the TCVGH database. After excluding 2295 participants due to missing data, the final dataset comprised 54,962 participants in the TCVGH database. Among these, 191, 135, and 110 patients were diagnosed with SCC of the esophagus, oropharynx, and hypopharynx, respectively. Furthermore, among these patients, 7 patients had SCC of the esophagus and oropharynx; 16 patients had SCC of the esophagus and hypopharynx, 20 patients had SCC of the oropharynx and hypopharynx, and 4 patients had SCC of the esophagus, oropharynx, and hypopharynx.

To further investigate the relationship between the PRS and the study outcome using Cox proportional hazard regression models, adjusted hazard ratios were calculated for SCC risk factors and polygenic risk scores in relation to the occurrence of first-onset SCC outcomes. The baseline characteristics of participants, along with HRs for SCC, adjusted for SCC risk factors, are shown in Table 1. Both PRSs were normally distributed (Appendix A), and PGS001087 exhibited an association with SCC outcome, showing HRs of 2.990 (95% CI = 1.858–4.811, *p* < 0.001) after adjustment for potential confounders. Similarly, PGS001394 showed an association with SCC outcome, with HRs of 2.421 (95% CI = 2.270–2.693, *p* < 0.001).

Furthermore, we explored the association between various PRSs and the risk of squamous cell carcinoma (SCC) using Cox regression analysis. The analysis was adjusted for age and gender, as presented in Appendix A. Polygenic scores related to tobacco consumption, such as PGS001046 or PGS001047, as well as PGS002063 (related to esophagitis, GERD, and related diseases), did not show any associations with SCC after adjusting for age and gender.

In Table 2, the study population was divided into various subgroups based on the specific SCCs that they exhibited. These subgroups included SCCs of the esophagus, oropharynx, and hypopharynx individually, as well as patients with a combination of two of these SCC types, and those with all three SCCs under investigation. Notably, when examining patients with SCC of the esophagus alone, the incidence rates showed an upward trend as the polygenic risk score (PRS) quartiles increased. These rates were 0.22%, 0.29%, 0.33%, and 0.55% in quartiles Q1 to Q4 with PGS001087, respectively. This trend was statistically significant, with a *p*-value of less than 0.001, indicating that as the PRS increased, the likelihood of SCC of the esophagus also increased. Similar results were observed in patients with SCC of the esophagus when stratified by quartiles with PGS001394, with incidence rates ranging from 0.17% to 0.60% across quartiles Q1 to Q4 (*p* < 0.001) (Appendix A). In addition, a similar trend was observed in patients with SCC of the hypopharynx alone, with the highest incidence occurring in quartile Q4, presenting rates of 0.15%, 0.22%, 0.15%, and 0.29% in quartiles Q1 to Q4 with PGS001087, respectively, reaching statistical significance with a *p*-value of 0.018 (Table 2). However, the analysis for patients with combined SCCs could not yield a meaningful *p*-value, likely due to the limited number of cases within this subgroup. This lack of statistical significance underscores the challenge of drawing definitive conclusions in situations with small sample sizes. These results revealed significant associations between PRS quartiles and the incidence of SCC of the esophagus alone and SCC of the hypopharynx alone.

The average scores for PGS001087 for different SCCs are presented in Table 3. The score range for PGS0001087 spans from a minimum of −0.7417 to a maximum of 0.8480, with a mean score of −0.0680. The scale check is 0.2052; the distribution is summarized by the lower quartile at −0.2133, the median at −0.0780, and the upper quartile at 0.0697. The patients with SCC of the esophagus and hypopharynx have the highest mean PRS (0.046 ± 0.2275), and the second-highest PRS was in the patients with SCC of the esophagus alone (0.0032 ± 0.217).

The study comprehensively assessed a range of clinical parameters, including age, gender, medical history, cigarette and alcohol use, and SCC levels, as detailed in Table 4. Notably, when analyzing patients with any of the three SCC subtypes (esophagus, oropharynx, or hypopharynx), no statistically significant correlations were identified between these clinical parameters and the quartiles of PRS. Additionally, the investigation delved into clinical staging, stratifying the SCCs into stages I to IV, as illustrated in Table 5. Remarkably, this analysis also yielded results showing no substantial correlation between the clinical staging of SCCs in these specific anatomical locations (esophagus, oropharynx, and hypopharynx) and the PRS quartiles. However, it is noteworthy that within the subset of patients with more advanced disease stages, there was a tentative increase in the incidence of these cancers with higher PRS quartiles, suggesting a potential association between genetic risk and disease severity in these cases. Furthermore, Table 6 presents the prognosis data for participants diagnosed with SCCs of the esophagus, oropharynx, or hypopharynx. In this analysis, no significant correlations were detected between the rates of recurrence and mortality and the PRS quartiles. These findings suggest that the clinical parameters and clinical stagings did not exhibit strong associations with PRS quartiles, although there was a potential trend of increased incidence among patients with advanced-stage SCCs in relation to higher PRS quartiles. Additionally, the PRS did not appear to significantly impact the rates of recurrence or mortality for patients with these SCCs.

## 4. Discussion

Our study aimed to utilize the PGS001087, with the phenotype of alcohol intake frequency, to investigate the association between disease-associated variants identified in GWAS and susceptibility to esophageal, oropharyngeal, and hypopharyngeal cancers within the TCVGH-TPMI cohort. We found that patients in the top PRS quartile had a significantly higher risk of SCC of the esophagus compared to those in the bottom quartile. However, although we found a higher mean PRS for SCCs of the esophagus and hypopharynx combined, we did not find a positive correlation between the PRS and a combination of the three cancers. Additionally, the PRS was not statistically correlated to the clinical stagings at diagnosis of the three cancers.

The PRS used in our study, PGS001087, was developed by Tanigawa et al. in a trans-ancestry study that included GWAS data from more than 269,000 individuals. Tanigawa et al. identified 813 sparse PRS models with significant incremental predictive performance when compared against the covariate-only model that considers age, sex, types of genotyping arrays, and the principal component loadings of genotypes [20]. Moreover, Tanigawa’s PRS has played a role in several cohort studies with positive results. Nachmanson et al. found that in patients with breast ductal carcinoma, the tissue-derived PRS was significantly associated with breast cancer subsequent events (HR = 2, 95% CI 1.2–3.8) [24]. Our study represents the first GWAS cohort to examine Tanigawa’s PRS in patients with SCCs of the upper gastrointestinal tract.

Comparing PGS0087187 to the current published PRSs for esophageal cancer, PGS003388 [25] was derived from a discovery analysis of 356,743 variants associated with esophageal adenocarcinoma, which has proved its prediction accuracy in European ancestry [25]. Similarly, PGS000363 [26], is associated with all esophageal cancers and shares similarities with PGS003388. It encompasses a substantial number of variants (1,081,646) and exhibits prediction accuracy primarily in European ancestry populations, potentially limiting its generalizability to other populations [26]. However, having too many variants in a PRS model can lead to complexity, overfitting, challenges in variant selection, increased data requirements, and uncertainty about the effects of gene mutations. Conversely, PGS002298 [27], which originated from a discovery analysis of 14 variants associated with esophageal cancer, is specifically applicable to individuals of European ancestry. However, it is worth noting that its predictive accuracy is relatively lower, with an AUROC of 0.53 [28].

The incidence of esophagus, oropharynx, or hypopharynx cancer is relatively lower in Taiwan compared to the cancers which have been better studied in PRS-related studies, such as lung, breast, and colon cancers [2]. The discrepancy may be attributed to the lack of systemic PRS screening; however, SCCs of the esophagus, oropharynx, and hypopharynx are considered highly related to each other, and, to the best of our knowledge, there were no previous PRS-related studies investigating the correlation between the PRS and the combination of the three SCCs.

There were still several limitations to our study. As it was a single-centered study, our sample size was relatively small, which has prevented the prediction of the risk of the combined cancers. Given the retrospective nature of our study and the limited sample size, we chose non-parametric tests, including the Chi-square test and Fisher’s exact test, to analyze categorical variables. Furthermore, the lack of recording of the clinical stagings has limited the accuracy of the correlation between the stages and the quartiles of the PRS. A similar lack related to retrospective data collection might also occur in the self-reports, as patients who drink less or only socially might unintentionally not have fully reported their drinking habits, which possibly led to the underestimation of the influence of alcohol consumption on PRS scores. Moreover, the self-reports only provided “yes” or “no” information, so the correlation between the amount of the substance used and the PRS was still uncertain. Lastly, the follow-up period was not long enough, which may have limited the generalizability of our findings in terms of disease treatment and prognosis.

Our study is the first GWAS cohort to examine this PRS (PGS001087) in patients with SCCs of the upper gastrointestinal tract. For patients with known risks or a family history of esophageal or hypopharyngeal cancer, it can be suggested as a reliable tool for incidence prediction, to encourage them to undergo early examinations or make lifestyle adjustments. Furthermore, routine gastroscopy examination may be applicable for high-risk patients with a high PRS score. The frequency of gastroscopy examinations may become a valuable issue in future research. Ultimately, a most feasible PRS and guidelines for regular follow-up for high-PRS patients in the Taiwanese population may be found and established after the expansion of the database, achieving the national goal of effective cancer prediction and early treatment. 

## 5. Conclusions

Our findings revealed a significant association between PGS001087 and the risk of SCCs. Individuals in the highest quartile of PGS001087 demonstrated a significantly increased susceptibility to esophageal and hypopharyngeal SCC compared to those in the lowest quartile. While there were no positive correlations between PGS001087 and the combination of the three cancers, nor with clinical cancer staging, this study still contributes to our understanding of genetic factors in upper aerodigestive tract cancers, particularly esophageal SCC, and provides valuable insights for future research and risk assessment. 

## Figures and Tables

**Table 1 cancers-16-00707-t001:** Risk of squamous cell carcinoma (SCC) risk factors and polygenic risk scores for first-onset SCC.

Squamous Cell Carcinoma (SCC) Risk Factors	HRs (95% CI)	*p* Value ^a^
Age, years	1.026 (1.019–1.033)	<0.001
Sex, male	7.681 (5.748–10.263)	<0.001
History of hypertension	0.398 (0.309–0.513)	<0.001
History of diabetes mellitus	0.511 (0.380–0.686)	<0.001
Cigarette smoking	9.185 (7.296–11.562)	<0.001
Alcohol drinking	1.208 (0.754–1.934)	0.432
Betel nut chewing	10.202 (8.313–12.519)	<0.001
Polygenic risk score (PRS)		
PGS001087 (Alcohol intake frequency)	2.990 (1.858–4.811)	<0.001
PGS001394 (Frequency of drinking alcohol)	2.421 (2.270–2.693)	<0.001

^a^ HRs were estimated using a Cox proportional hazards model, stratified by sex, and adjusted for age at baseline, smoking status, alcohol drinking, betel nut chewing, history of diabetes, and history of hypertension, where appropriate. Continuous variables are presented as HRs for each standard deviation higher than each predictor to facilitate comparison. Categorical variables include HRs for men versus women, patients with diseases versus without, current smokers versus others, drinking versus others, and betel nut chewing versus others.

**Table 2 cancers-16-00707-t002:** Classification of the study population with the three squamous cell carcinomas (SCCs).

Variables (SCC)	Quartiles of the Polygenic Risk Score (PGS001087)								*p* Value ^a^
	Q1 (*n* = 13,740)		Q2 (*n* = 13,736)		Q3 (*n* = 13,742)		Q4 (*n* = 13,744)		
	N	%	N	%	N	%	N	%	
Esophagus									<0.001
No	13,710	99.78	13,696	99.7	13,697	99.7	13,668	9.5	
Yes	30	0.22	40	0.29	45	0.33	76	0.55	
Oropharynx									0.822
No	13,705	99.75	13,701	99.75	13,713	99.79	13,708	99.74	
Yes	35	0.25	35	0.25	29	0.21	36	0.26	
Hypopharynx									0.018
No	13,720	99.85	13,706	99.78	13,722	99.85	13,704	99.71	
Yes	20	0.15	30	0.22	20	0.15	40	0.29	
Esophagus + oropharynx									-
No	13,738	99.99	13,732	99.97	13,739	99.98	13,742	99.99	
Yes	2	0.01	4	0.03	3	0.02	2	0.01	
Esophagus + hypopharynx									-
No	13,738	99.99	13,731	99.96	13,739	99.98	13,734	99.93	
Yes	2	0.01	5	0.04	3	0.02	10	0.07	
Oropharynx + hypopharynx									-
No	13,734	99.96	13,727	99.93	13,737	99.96	13,740	99.97	
Yes	6	0.04	9	0.07	5	0.04	4	0.03	
Esophagus + oropharynx + hypopharynx									-
No	13,740	100.00	13,734	99.99	13,741	99.99	13,743	99.99	
Yes	0	0.00	2	0.01	1	0.01	1	0.01	

^a^ Categorical variables were expressed as numbers (percent) and were analyzed using the Chi-square test.

**Table 3 cancers-16-00707-t003:** Clinical relevance: mean polygenic risk scores among the study participants.

Variables (SCC) ^a^	Esophagus (*n* = 164)		Oropharynx (*n* = 104)		Hypopharynx (*n* = 70)		Eso ^b^ + Oro ^c^ (*n* = 7)		Eso + Hyp ^d^ (*n* = 16)		Oro + Hyp (*n* = 20)		Eso + Oro + Hyp (*n* = 4)		*p* Value ^f^
	N	%	N	%	N	%	N	%	N	%	N	%	N	%	
Demography															
Age (mean/SD) ^e^	66.07	10.81	63.99	10.46	63.59	8.54	61.14	12.01	62.31	9.46	64.45	10.2	56.75	7.8	0.216
Onset age (mean/SD) ^e^	60.11	10.45	57.08	10.62	57.83	8.53	54	13.6	55.31	10.66	56.45	11.13	50.5	4.04	0.058
Gender (*n*, %)															0.174 ^‡^
female	29	17.58	17	16.35	5	7.14	0	0	0	0	2	10	0	0	
male	135	82.32	87	83.65	65	92.86	7	100	16	100	18	90	4	100	
PGS001087															
Score (mean/SD) ^e^	0.0032	0.217	−0.0521	0.222	−0.0138	0.2139	−0.0921	0.2824	0.0462	0.2275	−0.1083	0.17	−0.0246	0.1665	0.138
PGS001394															
Score (mean/SD) ^e^	0.1774	0.0836	0.1525	0.0749	0.1624	0.0843	0.1585	0.0444	0.2045	0.0894	0.155	0.0544	0.2154	0.0763	0.0672

^a^ SCC: Squamous cell carcinoma. ^b^ Eso: Squamous cell carcinoma of the esophagus. ^c^ Oro: Squamous cell carcinoma of the oropharynx. ^d^ Hyp: Squamous cell carcinoma of the hypopharynx. ^e^ Continuous variables were expressed as mean ± standard deviation (SD) and were analyzed using Student’s *t*-test for normal data distributions. ^f^ Categorical variables were expressed as numbers (percent) and were analyzed using the Chi-square test. **^‡^** Categorical variables were expressed as numbers (percent) and were analyzed using Fisher’s exact probability test.

**Table 4 cancers-16-00707-t004:** Characteristics of the study subjects (in patients with any of the three cancers).

Variables	Quartiles of the Polygenic Risk Score (PGS001087)								*p* Value ^b^
	Q1 (*n* = 97)		Q2 (*n* = 96)		Q3 (*n* = 96)		Q4 (*n* = 96)		
	N	%	N	%	N	%	N	%	
Demography									
Age (mean/SD) ^a^	65.18	10.85	64.93	9.85	64.49	11.21	63.92	9.2	0.843
Onset age (mean/SD) ^a^	58.24	11.22	58.17	10.04	58.44	11.27	58.26	8.72	0.998
Gender (*n*, %)									0.808
female	12	12.37	13	13.54	16	16.67	12	12.5	
Male		87.63	83	86.46	80	83.33	84	87.5	
Medical history (*n*, %)									
Family history	1	25	1	12.5	2	50	2	50	0.491 ^‡^
Hypertension									0.958
No	78	80.41	76	79.17	79	82.29	77	80.21	
Yes	19	19.59	20	20.83	17	17.71	19	19.79	
Diabetes mellitus									0.633
No	85	87.63	80	83.33	83	84.46	86	89.58	
Yes	12	12.37	16	16.67	13	13.54	10	10.42	
Other malignancy (*n*, %)									0.317
No	53	54.64	59	61.46	65	67.71	60	62.5	
Yes	44	45.36	37	38.54	31	32.29	36	37.5	
Lifestyle (*n*, %)									
Cigarette smoking	71	73.2	73	76.04	70	72.92	74	77.08	0.883
Alcohol drinking	6	9.38	2	3.39	5	10	6	13.64	0.282 ^‡^
Betel nut chewing	61	62.89	59	61.46	62	64.58	52	54.17	0.468
Laboratory data (mean/SD) ^a^									
Squamous cell carcinoma antigen (SCC)	1.44	1.25	1.36	0.88	1.62	2.1	1.59	1.55	0.756

^a^ Continuous variables were expressed as mean ± standard deviation (SD) and were analyzed using Student’s *t*-test for normal data distributions. ^b^ Categorical variables were expressed as numbers (percent) and were analyzed using the Chi-square test. **^‡^** Categorical variables were expressed as numbers (percent) and were analyzed using Fisher’s exact probability test.

**Table 5 cancers-16-00707-t005:** Clinical staging analysis of the three SCCs ^a^ from Stage I to Stage IV.

Variables	Quartiles of the Polygenic Risk Score (PGS001087)								*p* Value ^b^
SCC of esophagus Clinical staging (*n*, %)	Q1 (*n* = 40)		Q2 (*n* = 48)		Q3 (*n* = 51)		Q4 (*n* = 52)		0.953
Stage I	3	21.43	1	5	3	11.11	3	9.09	
Stage II	4	28.57	5	25	5	18.52	6	18.18	
Stage III	4	28.57	10	50	14	51.85	14	42.42	
Stage IV	2	14.29	2	10	3	11.11	6	18.18	
SCC of oropharynx Clinical staging (*n*, %)	Q1 (*n* = 42)		Q2 (*n* = 36)		Q3 (*n* = 31)		Q4 (*n* = 26)		0.938
Stage I	4	30.77	4	25	3	23.08	3	23.08	
Stage II	2	15.38	0	0	1	7.69	2	15.38	
Stage III	1	7.69	3	18.75	2	15.38	2	15.38	
Stage IV	6	46.15	9	56.25	7	53.85	6	46.15	
SCC of hypopharynx Clinical staging (*n*, %)	Q1 (*n* = 28)		Q2 (*n* = 30)		Q3 (*n* = 24)		Q4 (*n* = 28)		0.626
Stage I	2	11.76	5	26.32	3	17.65	2	11.11	
Stage II	2	11.76	1	5.26	1	5.88	4	22.22	
Stage III	1	5.88	4	21.05	2	11.76	4	22.22	
Stage IV	11	64.71	9	47.37	11	64.71	8	44.44	

^a^ SCCs: Squamous cell carcinomas. ^b^ Categorical variables were expressed as numbers (percent) and were analyzed using Fisher’s exact probability test.

**Table 6 cancers-16-00707-t006:** Correlation analysis of recurrence and mortality rates with the PRS ^a^ quartiles in study participants.

Variables	Quartile of the Polygenic Risk Score (PGS001087)								*p* Value ^b^
	Q1 (*n* = 40)		Q2 (*n* = 48)		Q3 (*n* = 51)		Q4 (*n* = 52)		
	N	%	N	%	N	%	N	%	
Outcome									
Recurrence	20	20.62	22	22.92	17	17.71	27	28.13	0.359
Mortality	20	20.62	18	18.75	12	12.5	15	15.63	0.453

^a^ PRS: Polygenic risk score. ^b^ Categorical variables were expressed as numbers (percent) and were analyzed using the Chi-square test.

## Data Availability

The datasets generated and/or analyzed during the current study are available from the corresponding author upon reasonable request, subject to approval by the Institutional Review Board.

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
