# Peer review of "Polygenic Risk Score in Predicting Esophageal, Oropharyngeal, and Hypopharynx Cancer Risk among Taiwanese Population"

_cancers, 2024, doi:10.3390/cancers16040707_

Round 1

Reviewer 1 Report

Comments and Suggestions for Authors

The study aims to investigate the association between the PRS and the SCCs of esophagus, oropharynx, and hypopharynx susceptibility, as well as to analyze the impact of genomewide susceptibility variants on clinical outcomes and prognosis.

The study covers some issues that have been overlooked in other similar topics. The structure of the manuscript appears adequate and well divided in the sections. Moreover, the study is easy to follow, but few issues should be improved. Some of the comments that would improve the overall quality of the study are:

a. Authors must pay attention to the technical terms acronyms they used in the text.

b. Conclusion Section: This paragraph required a general revision to eliminate redundant sentences and to add some "take-home message".

Author Response

Author’s Response to reviewer’s comments

Reviewer #2

Comments and Suggestions for Authors

The study aims to investigate the association between the PRS and the SCCs of esophagus, oropharynx, and hypopharynx susceptibility, as well as to analyze the impact of genomewide susceptibility variants on clinical outcomes and prognosis.

The study covers some issues that have been overlooked in other similar topics. The structure of the manuscript appears adequate and well divided in the sections. Moreover, the study is easy to follow, but few issues should be improved. Some of the comments that would improve the overall quality of the study are:

  1. Authors must pay attention to the technical terms acronyms they used in the text.

Author’s response

Thank you for your advice. We already completed the lacking notes of abbreviations in the manuscript. We also listed all the acronyms we used in the manuscript for the convenience of the readers (Please see page 10, abbreviations section).

  1. Conclusion Section: This paragraph required a general revision to eliminate redundant sentences and to add some "take-home message".

Author’s response

Thank you for your constructive advice. Redundant sentences have been deleted from the conclusion section, and the key points of this study have also been highlighted (Please see page 10, conclusions section).

Reviewer 2 Report

Comments and Suggestions for Authors

To investigate the strong association between tube cancer and oropharyngeal and hypopharyngeal cancers, the authors conducted a retrospective study of 54,692 participants, including 385 patients with esophageal, oropharyngeal, or hypopharyngeal squamous cell carcinoma, at Taichung Veterans General Hospital. This research is of great value. The article is generally written clearly and logically. Here are a few small issues to be aware of.

1. Regarding the selection of statistical methods, the author can briefly explain the reasons for using different statistical methods to calculate the p-value.

2. How the authors' findings guide future research and risk assessment strategies can be briefly explained.

Author Response

Author’s Response to reviewer’s comments

Reviewer #3

Comments and Suggestions for Authors

To investigate the strong association between tube cancer and oropharyngeal and hypopharyngeal cancers, the authors conducted a retrospective study of 54,692 participants, including 385 patients with esophageal, oropharyngeal, or hypopharyngeal squamous cell carcinoma, at Taichung Veterans General Hospital. This research is of great value. The article is generally written clearly and logically. Here are a few small issues to be aware of.

  1. Regarding the selection of statistical methods, the author can briefly explain the reasons for using different statistical methods to calculate the p-value.

Author’s response

Thank you for the valuable insights provided by the reviewer. In this study, different statistical methods were employed based on the nature of the variables and the characteristics of the data. For categorical variables, we utilized the Chi-square test. In instances where the minimum expected count was less than 5, Fisher's Exact Test was employed to assess group differences more accurately.

Each analysis table clearly specifies the statistical method used, and corresponding annotations are provided below the tables. This ensures that readers have a clear understanding of the statistical analysis methods employed and the rationale behind their selection. Please see the revised manuscript for details (Please see page 4; Table 2 & Table 3).

  1. How the authors' findings guide future research and risk assessment strategies can be briefly explained.

Author’s response

Thank you for your visionary question about the future research and risk assessment strategies. We will recommend the patients with known risks or family history of esophageal or hypopharyngeal cancer to check PRS score. Furthermore, we will suggest lifestyle adjustment and routine gastroscopy examination for high-risk patients with a high PRS score. The frequency of gastroscopy exam may become a valuable issue of future research. We have added the above opinion to the discussion section in the revised manuscript (Please see page 9, last paragraph of the discussion section).

Reviewer 3 Report

Comments and Suggestions for Authors

I have carefully reviewed your manuscript entitled Polygenic Risk Score in Predicting Esophageal, Oropharyngeal, and Hypopharynx Cancer Risk Among Taiwanese Population submitted to Cancers. I appreciate the effort and time you invested in your research and I find that the topic is of great significance. Overall, I believe that your work has the potential to make a valuable contribution to the scientific community. However, several areas that require attention have been identified before the manuscript can be considered for publication.

-          Polymorphism measurement

The authors must justify why, out of all the polymorphisms associated with alcohol consumption, they have only used the PGS001087 PRS. Specifically, considering that there are 29 polymorphisms associated solely with alcohol, PGS001394 is also related to the Asian population. Furthermore, the authors should justify why they have not used the polymorphisms and their association with tobacco consumption, PGS001046 or PGS001047.

-          Incidence

Given the retrospective nature of the study, it is not specified in the study methodology whether they are incident cases and whether the assessment of consumption has a follow-up prior to the cancer diagnosis. Please clarify or correct this.

Statistical Analysis

In statistical analysis, the authors need to make the most modifications. I believe it has several shortcomings. First, the authors do not assess the normality assumption for using parametric tests, which I doubt can be applied to sample sizes of N=7, including some subgroups. The use of univariate logistic regression for outcome analysis is not appropriate; it should be a multivariate Cox regression. Finally, it should be noted that in epidemiological studies that evaluate exposure to consumption or environmental exposures and genetic components of individuals, the preferred techniques are Manhattan plot analyses.

Results

Regarding the results, I have one issue that caught my attention, and the authors should clarify them. The most striking is that they are assessing a polymorphism related to alcohol consumption; however, patients with cancer have an extremely low level of alcohol exposure. Only 19 of 436 patients consume alcohol and this frequency is so low that it does not even correspond to a risk factor for this type of cancer.

Author Response

Author’s Response to reviewer’s comments

Reviewer #4

Comments and Suggestions for Authors

I have carefully reviewed your manuscript entitled Polygenic Risk Score in Predicting Esophageal, Oropharyngeal, and Hypopharynx Cancer Risk Among Taiwanese Population submitted to Cancers. I appreciate the effort and time you invested in your research and I find that the topic is of great significance. Overall, I believe that your work has the potential to make a valuable contribution to the scientific community. However, several areas that require attention have been identified before the manuscript can be considered for publication.

-          Polymorphism measurement

The authors must justify why, out of all the polymorphisms associated with alcohol consumption, they have only used the PGS001087 PRS. Specifically, considering that there are 29 polymorphisms associated solely with alcohol, PGS001394 is also related to the Asian population. Furthermore, the authors should justify why they have not used the polymorphisms and their association with tobacco consumption, PGS001046 or PGS001047.

Author’s response

Thank you for your valuable comment regarding the use of PGS001087 PRS in our study. To investigate the relationship between PRS and the study outcome, in addition to PGS001087 PRS, we explored various PGS in our study. As shown in Supplementary Table 1, in addition to PGS001087, we have included supplementary tables that list different PGS, including PGS001394 (Frequency of drinking alcohol), PGS002291 (Nasopharyngeal carcinoma), PGS002063 (Esophagitis, GERD, and related diseases), PGS001046 (Past tobacco smoking – Smoked at least once), and PGS001047 (Past tobacco smoking – Smoked at least once), sourced from the PGS catalog. When comparing the highest quartile (Q4) to the lowest quartile (Q1), the OR for SCC risk was 2.01 (95% CI=1.50-2.71, p<0.001), signifying a statistically significant association. Similarly, when comparing Q3 to Q1, the OR for SCC risk was 1.38 (95% CI=1.01-1.89, p=0.0431). In addition to PGS001087, we explored the relationship between various polygenic scores and the study outcome. Notably, PGS001394 (Frequency of drinking alcohol) revealed a significantly higher risk of SCC in Q4 compared to Q1 (OR=1.83, 95% CI=1.38-2.44, p<0.001). In contrast, we did not observe a significant association between tobacco consumption-related polygenic scores, such as PGS001046 or PGS001047, and squamous cell carcinomas in our study. (Please see page 4, first paragraph of the results section; Supplementary Table 1).

-          Incidence

Given the retrospective nature of the study, it is not specified in the study methodology whether they are incident cases and whether the assessment of consumption has a follow-up prior to the cancer diagnosis. Please clarify or correct this.

Author’s response

Thank you for your visionary question and opinion. This is a retrospective study, in which all participants were outpatients followed up in Taichung Veterans General Hospital (TCVGH). From the perspective of research methodology, we utilized the database of TCVGH and the standardized diagnostic codes to distinguish the diseased patients from the healthy majority of all the participants and to perform statistical analyses. It is important to note that the patients in this study were not asymptomatic latent individuals (incident cases). Therefore, there have been no clinical recommendations for continued follow-up made for participants with high PRS scores yet. However, this aspect may serve as a valuable direction for future research endeavors.

Statistical Analysis

In statistical analysis, the authors need to make the most modifications. I believe it has several shortcomings. First, the authors do not assess the normality assumption for using parametric tests, which I doubt can be applied to sample sizes of N=7, including some subgroups. The use of univariate logistic regression for outcome analysis is not appropriate; it should be a multivariate Cox regression. Finally, it should be noted that in epidemiological studies that evaluate exposure to consumption or environmental exposures and genetic components of individuals, the preferred techniques are Manhattan plot analyses.

Author’s response

Thank you for your comments. In this study, our statistical analysis did not use logistic regression. Due to the small sample sizes, we opted for the Chi-square test and, in cases where the minimum expected count was less than 5, we utilized Fisher's Exact Test. We acknowledge the limitations associated with small sample sizes and appreciate your suggestions. Please see the revised manuscript for details (Please see page 4; Table 2 & Table 3).

Regarding the use of Manhattan plot analyses, we understand its relevance in genomics studies. However, for our specific focus on the association between polygenic risk scores (PRS) and susceptibility to esophageal and hypopharyngeal cancer, we found that Chi-square test and Fisher's Exact Test were deemed appropriate statistical methods to address our research questions.

Results

Regarding the results, I have one issue that caught my attention, and the authors should clarify them. The most striking is that they are assessing a polymorphism related to alcohol consumption; however, patients with cancer have an extremely low level of alcohol exposure. Only 19 of 436 patients consume alcohol and this frequency is so low that it does not even correspond to a risk factor for this type of cancer.

Author’s response

Thank you for the valuable comment regarding alcohol consumption. We admit that this was precisely one of the limitations of our study. The lack of the number of patients with alcohol consumption might occur while history taking that patients with lower amount of alcohol consumption or social drinking might tend to deny history of alcohol drinking unintentionally, which possibly led to the underestimation of the influence of alcohol consumption to the PRS scores. We have already added the above opinion as our limitation in the discussion section (Please see page 9).

Round 2

Reviewer 3 Report

Comments and Suggestions for Authors

The authors have neither responded to nor modified the previous comments sent for review. The response to the incident cases is unsatisfactory, and there is no modification of the statistical results that are not appropriate. They continue to use parametric tests despite not evaluating the fit to a normal distribution (Descriptive statistics for continuous variables were presented as mean ± standard desviation (SD), and group differences were assessed using Student's t-test) They use logistic regression when the most suitable technique is Cox regression (To further explore the relationship between PRS and the study outcome, univariable logistic regression analysis was conducted to calculate the odds ratio (OR) and 95% confidence interval (95% CI)) and they have a significant limitation regarding the number of individuals with alcohol consumption, raising doubts about whether the results demonstrate an unequivocal association or are conditioned by other factors.

Author Response

Reviewer Comments:

Reviewer 3

Comments and Suggestions for Authors

The authors have neither responded to nor modified the previous comments sent for review. The response to the incident cases is unsatisfactory, and there is no modification of the statistical results that are not appropriate. They continue to use parametric tests despite not evaluating the fit to a normal distribution (Descriptive statistics for continuous variables were presented as mean ± standard desviation (SD), and group differences were assessed using Student's t-test) They use logistic regression when the most suitable technique is Cox regression (To further explore the relationship between PRS and the study outcome, univariable logistic regression analysis was conducted to calculate the odds ratio (OR) and 95% confidence interval (95% CI)) and they have a significant limitation regarding the number of individuals with alcohol consumption, raising doubts about whether the results demonstrate an unequivocal association or are conditioned by other factors.

Author’s response

Thank you for your valuable feedback on our manuscript. We appreciate the time and effort you have dedicated to reviewing our work. We have carefully considered your comments and would like to address some of the concerns you raised.

Thank you for your thoughtful consideration of the statistical methods used in our study. In our study, the PRS was normally distributed, as shown below:

Supplementary Figure 1

Supplementary Figure 1. Distribution of PGS001087 (A) and PGS001394 (B) by incident SCC.

In this study, we utilized the PGS001087 PRS and PGS001394, which were derived from a discovery analysis of variants associated with alcohol consumption measurement identified from a Trans-ancestry genome-wide association study (GWAS) for 8353 and 860 curated traits, respectively, across nine ancestry groups from the United Kingdom Biobank.

The suggestion to consider Cox regression over logistic regression has been duly noted, and we thank the reviewer for their insightful input. In response, we have employed Cox proportional hazards models to analyze the hazard ratios for the study outcome. These models were stratified by sex, with time since study entry serving as the timescale. Additionally, outcomes were censored appropriately, accounting for participants lost to follow-up, deceased participants, or reaching the end of available follow-up (November 2022). As detailed in Table 1, we present the baseline characteristics of participants alongside hazard ratios (HRs) for SCC, adjusted for SCC risk factors.  Upon analysis, PGS001087 displayed a significant association with SCC outcome, yielding HRs of 2.990 (95% CI=1.858-4.811, p<0.001) after adjusting for potential confounders. Similarly, PGS001394 exhibited a significant association with SCC outcome, demonstrating HRs of 2.421 (95% CI=2.270-2.693, p<0.001) (Please see page 4, second paragraph of the results section; Table 1).

Table 1. Risk of squamous cell carcinoma (SCC) risk factors and polygenic risk scores for first-onset SCC outcomes.

Squamous cell carcinoma (SCC) risk factors

HRs (95% CI)

P valuea

Age, years

1.026 (1.019-1.033)

<0.001

Sex, male

7.681 (5.748-10.263)

<0.001

History of hypertension

0.398 (0.309-0.513)

<0.001

History of diabetes mellitus

0.511 (0.380-0.686)

<0.001

Cigarette smoking

9.185 (7.296-11.562)

<0.001

Alcohol drinking

1.208 (0.754-1.934)

0.432

Betel nut chewing

10.202 (8.313-12.519)

<0.001

Polygenic Risk Scores (PRS)

PGS001087 (Alcohol intake frequency)

2.990 (1.858-4.811)

<0.001

PGS001394 (Frequency of drinking alcohol)

2.421 (2.270-2.693)

<0.001

a HRs were estimated using Cox proportional hazards model, stratified by sex, and adjusted for age at baseline, smoking status, alcohol drinking, betel nut chewing, history of diabetes, and history of hypertension, where appropriate.

Regarding the incident cases, we carefully rechecked your question and the definition of an incident case. All participants in this study were incident cases and had not undergone regular follow-up prior to their cancer diagnoses. We have added the clarification to our manuscript in the method section (Please see page 4, method section).

Additionally, we acknowledge the limitation related to the number of individuals with alcohol consumption data and the challenges associated with self-reports and retrospective data collection. Unfortunately, due to the retrospective nature of the study, the proportion of patients providing information on alcohol consumption was limited. We explicitly discuss this limitation and its potential impact on the interpretation of our findings in the revised manuscript (Please see page 4, method section; page 10, limitation section).

It's important to note that our study faced challenges related to a small sample size, and we acknowledge that statistical test outcomes may be influenced by this limitation. Given the hospital-based retrospective nature of our observational study and the constrained sample size, we made a deliberate choice to employ non-parametric tests, specifically the Chi-square test and Fisher's Exact Test, for the analysis of categorical variables (Please see page 7; Table 3 & Table 4; page 10, limitation section).

We appreciate your insightful comments, and we are committed to improving the clarity and robustness of our manuscript in response to your suggestions. If you have any further recommendations or specific points you would like us to address, please feel free to let us know.

Round 3

Reviewer 3 Report

Comments and Suggestions for Authors

Dear authors.

I am writing to express my sincere appreciation for your dedication and meticulous efforts in addressing the suggestions and revisions outlined in the previous communication regarding your manuscript.

I am pleased to acknowledge that your revisions have been implemented effectively, and the manuscript now meets the standards expected for publication in our scientific journal.